# Effect of Interlayer Delay on the Microstructure and Mechanical Properties of Wire Arc Additive Manufactured Wall Structures

**DOI:** 10.3390/ma14154187

**Published:** 2021-07-27

**Authors:** Shalini Singh, Arackal Narayanan Jinoop, Gorlea Thrinadh Ananthvenkata Tarun Kumar, Iyamperumal Anand Palani, Christ Prakash Paul, Konda Gokuldoss Prashanth

**Affiliations:** 1Mechatronics and Instrumentation Lab, Discipline of Mechanical Engineering, Indian Institute of Technology Indore, Indore 453552, Madhya Pradesh, India; s13singh2013@gmail.com (S.S.); mtech1902103012@iiti.ac.in (G.T.A.T.K.); palaniia@iiti.ac.in (I.A.P.); 2Raja Ramanna Centre for Advanced Technology, Laser Technology Division, Indore 452013, Madhya Pradesh, India; anjinoop18@gmail.com; 3Department of Engineering Sciences, Homi Bhabha National Institute, Anushaktinagar, Mumbai 400094, Maharashtra, India; drcppaul2015@gmail.com; 4Department of Mechanical and Industrial Engineering, Tallinn University of Technology, 19086 Tallinn, Estonia; 5Erich Schmid Institute of Materials Science, Austrian Academy of Sciences, A-8700 Leoben, Austria; 6Centre for Biomaterials, Cellular and Molecular Theranostics (CBCMT), Vellore Institute of Technology, Vellore 632014, Tamil Nadu, India

**Keywords:** wire arc additive manufacturing, interlayer delay, characterization, stainless steel 316L

## Abstract

Wire arc additive manufacturing is a metal additive manufacturing technique that allows the fabrication of large size components at a high deposition rate. During wire arc additive manufacturing, multi-layer deposition results in heat accumulation, which raises the preheat temperature of the previously built layer. This causes process instabilities, resulting in deviations from the desired dimensions and variations in material properties. In the present study, a systematic investigation is carried out by varying the interlayer delay from 20 to 80 s during wire arc additive manufacturing deposition of the wall structure. The effect of the interlayer delay on the density, geometry, microstructure and mechanical properties is investigated. An improvement in density, reduction in wall width and wall height and grain refinement are observed with an increase in the interlayer delay. The grain refinement results in an improvement in the micro-hardness and compression strength of the wall structure. In order to understand the effect of interlayer delay on the temperature distribution, numerical simulation is carried out and it is observed that the preheat temperature reduced with an increase in interlayer delay resulting in variation in geometry, microstructure and mechanical properties. The study paves the direction for tailoring the properties of wire arc additive manufacturing-built wall structures by controlling the interlayer delay period.

## 1. Introduction

Metal Additive Manufacturing (MAM) is an advanced manufacturing technique for building complex three-dimensional components directly from 3D model data, which increases productivity and production flexibility over traditional manufacturing processes [1]. Among the various MAM techniques, directed energy deposition (DED) is preferred over powder bed fusion (PBF) for producing large-size near-net-shaped components with a higher build rate [1,2]. Wire arc additive manufacturing (WAAM), which is a DED method, is a droplet-based additive manufacturing approach that requires the melting of the wire by using an arc as the heat source. As compared to the PBF based techniques, WAAM can build components with high density permitting the fabrication of high-performance components [3,4]. Furthermore, the system and operating cost of WAAM is lower than laser or electron beam additive manufacturing (AM) systems as it employs a commercial welding machine. In addition, the cost of the metal wire used as feedstock is approximately 10% that of the powder [5]. This attracts wide deployment WAAM technique for various engineering applications.

Being a multi-physics phenomenon, there are several factors that controls the microstructure evolution during WAAM. Furthermore, it is critical to understand the co-relation between the microstructure and mechanical properties, which aids in process selection for various applications. The relation between the deposit’s thermal history and microstructure is being studied by several researchers. Jinguo et al. [6] studied the cooling rate and microstructure changes during cold metal transfer-WAAM of 2Cr13 stainless steel based on deposit height. It was discovered that the amount of martensite generated in each layer differed depending on the cooling rates of each layer. Asala et al. [7] employed Inconel 718 to study microstructure variation in relation to deposit height while measuring temperature history during WAAM. The lower and middle parts of the deposit stayed within the Inconel 718 ageing temperature range for a longer amount of time during deposition than the higher part, resulting in increased precipitation hardening in the lower and middle parts. In order to regulate the mechanical properties, the deposit’s thermal parameters must be modelled for cost savings [8,9]. Few research have attempted to comprehend the impact of WAAM’s thermal history. Hejripour et al. [10,11] used the WAAM process to create wall-shaped and tube-shaped deposits out of 2209 duplex stainless steel (DSS) and created 3D numerical thermal models to examine the temperature history and cooling rate of each deposited layer. When the cooling rate of each deposited layer was less than 50 °C/s, the ferrite to austenite transformation increased. Jun et al. [12] used WAAM to generate a 3D transient heat transport model from a circular wall deposit made of mild steel. During deposition, the greatest temperature gradient for each layer was calculated when the heat source passes through the layer’s center. The thermal properties of the aforementioned investigation were replicated in a variety of settings.

During WAAM, multi-layer deposition results in heat accumulation, which raises the preheat temperature of the previously formed layer [13]. This causes process instabilities, resulting in deviations from the desired dimensions. Furthermore, the build-up of preheat temperature affects the cooling rate during WAAM deposition, which results in variations in metallurgical properties. By changing the temperature distribution during the process, the geometry and metallurgical properties of WAAM built structures may be tailored. Researchers have sought to regulate the temperature distribution and, therefore, the build features by varying the process parameters and process conditions [13]. According to Wang et al. [14], the distance between the trailing end and the center of the molten pool grew by 1.95 mm from the first to the fifth layer during the arc-wire deposition process due to increasing heat accumulation. Zhao et al. [15] previously published similar findings on the investigation of thermal behaviors during multi-layer deposition. Denlinger et al. [16] discovered that the inter-layer dwell duration, which is directly connected to the thermal properties, has a substantial impact on the distortion and residual stresses in as-fabricated titanium and nickel alloy components. Even though these modelling and experimental investigations have produced some valuable information, due to the complexity of the WAAM process, the underlying processes of arc characteristics and metal transfer behavior linked with heat accumulation remain poorly understood. As a result, one of the major challenges connected to the WAAM process is the dimensional deviations due to heat accumulation [13], as the thermal history largely impacts the geometry and microstructure, as seen in the literature. The heat accumulation rises with the number of layers and, thus, the characteristics of WAAM structures can be controlled by varying the interlayer temperature during the process by utilizing specified dwell durations or forced cooling [17]. The deviation in the dimension of the built structures is primarily dependent on the temperature accumulation during deposition, which in turn controls the outward melt-pool flow. The temperature accumulation can be controlled by adjusting the preheat temperature available on the previously deposited layer before laying on the next layer. The preheat temperature is primarily controlled by varying the process parameters or process conditions. Controlling the interlayer temperature can also help with oxidation control during deposition, especially when working with reactive materials [18]. To the best of the author’s knowledge, there is no public literature describing a systematic investigation on the influence of interlayer delay on the microstructure and mechanical properties of WAAM built stainless steel structures. Thus, the present study seeks to examine the impact of various interlayer delays on the geometry, microstructure and mechanical characteristics.

## 2. Materials and Methods

Stainless steel 316 L (SS 316L, Excel Metal & Engg Industries, Mumbai, India) with a wire diameter of 1.2 mm is used as the feedstock material. Table 1 presents the chemical composition of the SS 316L wire used for deposition. Gas Metal Arc Welding (GMAW) is used for the deposition of SS 316L wall.

The wire feeder (Esab, India) supplied the wire as a feedstock to the welding torch (Esab, India) connected to the X-Y stage. The X-Y stage is compiled with the electrode for movement in different directions. The X-Y controller is programed with G and M code, which is assisted by Repitier host software (V1.55, Willich, Germany) The process parameters are optimized considering continuity and uniform deposition of SS 316L single tracks. Table 2 shows the optimized parameter used for the deposition of the wall structures. In order to analyze the effect of interlayer delay, walls are built with an interlayer delay of 20, 40 and 80 s at optimized process parameters. Figure 1 presents the photographic view of the wall structures. The fabricated samples are systematically investigated using characterization techniques to examine the geometrical characteristics, microstructure and mechanical properties.

An one-color pyrometer Sensortherm METIS M318 (Sensortherm GmbH, Sulzbach, Germany with a temperature range and spectral range of 150 to 1200 °C and 1.65–2.1 μm, respectively) was used for measuring the temperature on the surface of the previously deposited layer at different interlayer delay conditions. The sectioning of the walls transverse to the laying direction is performed using a wire-cut electrical discharge machine (Hechang Machinery, Changzhou Hechang Intelligent Technology Co., Ltd., Changzhou, China). A Mettler balance (Type AE200, (Marshall Scientific, Hampton, NH, USA) with a density measurement instrument for solid materials (Type AB33360, (Marshall Scientific, Hampton, NH, USA) is used to determine density. In addition, samples are prepared according to standard metallographic procedures. An X-ray diffractometer (Make and model: BRUKER-D8 Advance, Bruker, Billerica, MA, USA) is utilized to perform X-ray diffraction studies (XRD) from 20° to 90° (step size: 0.02° and dwell time: 0.5 s) and the Scherrer formula is used to calculate crystallite size [19]. For microstructural examination, a scanning electron microscope (SEM) (Make and Model: S-4800 Hitachi, Tokyo, Japan) is employed. The secondary electron (SE) detector was used for obtaining the SEM images [20]. With a dwell period of 10 s and a load of 1.96 N, the microhardness is measured using a Vickers microhardness tester (Make and Model: WlterUhl-VMHT002, Newage, hardness testing, Horsham, PA, USA). For the compression test, a sample size (as per ASTM 9) with dimensions of 3 mm width and 6 mm length is employed.

## 3. Numerical Model

The ABAQUS 6.20 SOFTWARE (2020, Johnston, RI, USA) is used to carry out the numerical simulation for understanding the effect of interlayer delay on the temperature distribution of the WAAM process. In order to model layer deposition, the “element birth technique” is utilized [21]. In this technique, material properties are progressively switched from quiet to active values according to the deposition process as it requires specific elements activation techniques [21] in simulating material deposition. The basic principle is that quiet material has an extremely low thermal conductivity, hence filler elements in quiet state will experience a significant increase in temperature only if they are directly heated by the external power source. In this work, activation start and end temperature are set according to material liquidus and solidus temperatures, respectively [21].

For the purposes of the aforementioned, all of the wall elements are first disabled and then activated progressively by following the arc movement. The idle time method is based on two models, one for the simulation of heating (deposition) and the other for the simulation of cooling. The only difference between the two models in terms of mesh topology, material properties and boundary conditions is the heat source, which is exclusively used in the heating model. Without any idle time at the end of the heating operation, the deposition of the current layer is simulated. The end state of this simulation, which comprises the initial temperature field and the active/inactive state of the elements, determines the starting conditions for the cooling phase. The first element activation state is determined by the starting value of the (T_max_) variable. After that, the cooling simulation commences and models the work piece’s thermal behavior during the idle period following the deposition of the current layer. The defined idle time is then used as input for the heating simulation of the new layer. This is performed by adding an initial idle time to the heating simulation of the generic layer “k”, which corresponds to the expected value for layer “k − 1.”

The analysis uses the GOLDAK heat source model, which reliably predicts transient temperature distributions [22]. The Goldak model, which is depicted in Figure 2, specifies a heat generation per unit volume in a moving frame of reference. The *x*-axis is in the feed direction, the *y*-axis is in the arc aiming direction and the right-hand rule determines the *y*-axis. The element size in the wall was required to be less than 1 mm in X, 0.6 mm in Y and 0.8 mm in the Z direction based on the mesh sensitivity results. The best aspect ratio for the element size was found to be one, which resolved the convergence issue and provided accurate results. The selected mesh for the wall consisted of cubic elements, with 180, 5 and 16 elements in the X, Y and Z directions, respectively (as shown in Figure 2b). The mesh size was doubled in the X and Z directions in the bedplate to reduce the computational time. The mechanical and thermal properties of the material examined in this work and the numerical framework are followed from previously reported work [23,24]. Table 3 presents the material properties used for simulation.

This thermal model can be used to forecast thermal characteristics such as heat flow distribution and temperature distribution at each time step allowing asymmetries in heat distribution across the molten pool to be modeled. For positive and negative x semi-axes, two alternative power distribution functions are constructed, which allows asymmetries in heat distribution across the molten pool to be modeled. The power density distribution functions are shown in Equation (1). The semi-axes of two ellipsoids centered at the origin of the frame of reference, as shown in Figure 2, are the coefficients a_f_,_r_, *b* and *c*. Since the parameter a has a double subscript, separate values are utilized depending on the x sign (a_f_ for positive and a_r_ for negative), resulting in two distinct functions. The spatial region where the power density decreases are represented by an ellipsoid surface to 5% of its previous high value. The semi-axes of ellipsoids are usually adjusted according to the size of the molten pool [20]. The Goldak model, as previously indicated, does not take into account the true power distribution between filler and base metal. In GMAW, arc power is provided to the molten pool in two manners: direct transfer from the electric arc to the base metal and droplet enthalpy transfer of filler metal melting energy. The power utilized to melt filler metal accounts for nearly half of the overall arc power, according to previous study [22,25]. Equations (1) and (2) shows the power density distribution functions.
(1)$$q_{f}(x,\;y,\;z,\;t)=\frac{6\times \sqrt{3}\times Q\times f\_f}{a\times b\times c\times \pi \times \sqrt{\pi }}\times e^{\left[-3\times {\left(\frac{x}{a}\right)}^{2}+{\left(\frac{y}{b}\right)}^{2}+{\left(\frac{\varepsilon }{c\_f}\right)}^{2}\right]}$$
(2)$$q_{r}(x,\;y,\;z,\;t)=\frac{6\times \sqrt{3}\times Q\times f\_r}{a\times b\times c\times \pi \times \sqrt{\pi }}\times e^{\left[-3\times {\left(\frac{x}{a}\right)}^{2}+{\left(\frac{y}{b}\right)}^{2}+{\left(\frac{\varepsilon }{c\_r}\right)}^{2}\right]}$$
(3)Q=η×I×V

The product of welding current, welding voltage and arc efficiency yields Q, which is the heat input per unit time. The total energy input per unit time created by the heat source is calculated by integrating the two power density functions in spatial dimensions.

The boundary conditions used for modelling heat source are presented in Equations (4) and (5).
(4)$$q(x,\;y,\;z,\;t)=\{\begin{array}{c} q_{f}\;for\;z\geq z_{0}\\ q_{r}\;for\;z<z_{0} \end{array}$$
(5)$$\frac{f_{f}}{c_{f}}=\frac{f_{r}}{c_{r}}\;\mathrm{where}\;f_{f}+f_{r}=2$$

If the following condition is met, the words f_f,r_ are distribution factors with different values for the frontward and backward ellipsoids. Free convection boundary conditions were established on the top and bottom surfaces of the base plate, as well as the vertical surfaces of the wall. According to literature correlations [24,25,26,27], the convection coefficients for the top surface of the base plate, the bottom surface and the vertical surface of the wall were 8.5 W/m^2^ K, 4 W/m^2^ K and 12 W/m^2^ K, respectively. General radiation to the environment boundary condition was included, with the material emissivity set at 0.2. The starting temperatures for both the environment and the materials were set at 298.16 K.

## 4. Results and Discussion

### 4.1. Density Measurement

The density of WAAM samples is measured using Archimedes’ principle [28] and volume-based measurements performed with a hydraulic scale. The density measurement using deionized water show that the density of the wall structures is 93%, 96.5% and 99.5% for walls built with interlayer delay of 20, 40 and 80 s, respectively (refer to Figure 3). This can be due to a reduction in the melt-pool temperature with an increase in the interlayer delay resulting in reduced vaporization effects and melt-pool turbulence.

### 4.2. Effect of Interlayer Delay on Geometrical Dimensions

Figure 1 shows the WAAM structures deposited at different interlayer delay time. It indicates the presence of humps and spatters during WAAM deposition. In addition, it can be observed that the wall structure built with 20 s interlayer delay period is non-uniform due to the heat accumulation effect, while the wall structures built with 40 and 80 s interlayer delay are uniform. The measured values of wall heights and width are presented in Figure 4. It is observed that the wall height reduced by 13% with an increase in interlayer delay from 20 to 40 s and a 23% reduction is observed in wall height with an increase in the interlayer delay from 40 to 80 s. In the case of wall width, a reduction of 13% and 5% are observed with an increase in interlayer delay from 20 to 40 s and 40 to 80 s, respectively. The reduction in wall height with an increase in interlayer delay can be primarily due to a reduction in the preheat temperature on the surface of the previously built layer resulting in a reduction in the deposition efficiency. On the other side, the reduction in wall width with an increase in interlayer delay is mainly due to the reduction in the outward flow of the melt pool due to a reduction in the preheat temperature. Cracks and depression are also observed at the lower interlayer delay due to the heat accumulation effect and residual stresses. “Humping” and “undercutting” can be caused by fluid movement within the molten pool [29]. The driving force behind the movement is stated to be the Marangoni effect or the thermocapillary effect, in which fluid motion is induced by changes in surface tension [30]. Due to the arc’s oscillation frequency, ripples can be found throughout the tracks [31], as observed in Figure 1.

### 4.3. Microstructural Analysis

Figure 5 presents the microstructure of the built wall structures obtained using scanning electron microscopy. The temperature range of interest for stainless steel is 800 to 500 °C, as this is where the majority of the microstructural changes occur. This is the temperature range where austenite undergoes its solid-state transformation. As a result, in WAAM developed micrographs, the interlayer delay duration is critical. The dark gray region δ phase exhibits fine vermicular morphology within the light gray γ phase matrix. The second δ phase is embedded in the γ matrix and a large number of small holes (the darker region with circular structure) were also distributed in the matrix due to the high interlayer temperature because of low interlayer delay as shown in Figure 5a. A part of δ phase re-dissolves in γ phase and the retained δ phase exhibits vermicular morphology owing to the effects of subsequent thermal cycles. A large number of equiaxed grains and cellular dendrites were formed and grain size was increased for the lower interlayer delay sample (20 s). The ferrite exhibits reticular morphology within the austenitic. The walls built with higher thermal delay show finer grains as compared to walls built with lower thermal delay samples, which is mainly due to an increase in the cooling rate with an increase in delay time. With an increase in interlayer delay time, a reduction in the preheat temperature on the surface of the previously built layer is observed, which results in an increase in the thermal gradient and cooling rates [13,32].

### 4.4. XRD Results

Figure 6 presents the XRD pattern of the WAAM deposited SS 316L wall structures. It shows the presence of bimodal microstructure with both austenitic and ferritic phases coexisting at the same time. This is mainly due to non-equilibrium solidification during the WAAM process. The estimated values of crystallite size are 15.55, 12.27 and 8.55 nm for walls built with interlayer delay of 20, 40 and 80 s, respectively. The crystallite size is reduced with an increase in the interlayer delay mainly due to an increase in the cooling rate. It is observed that peak width increased with an increase in the interlayer delay, which is evident from the variation in the crystallite size. A magnified view of the peaks as observed in Figure 6 indicates a shift in the peak position, which can be due to variation in the lattice spacing with an interlayer delay. As the interlayer delay changes, the thermal gradient and thermal strain changes, which results in the shifting of peak position [33].

### 4.5. Mechanical Properties

Figure 7 presents the average micro-hardness of the WAAM built wall structures with different values of interlayer delay. It can be observed that the microhardness increased from 190 to 300 HV with an increase in interlayer delay. The higher hardness at 80 s is mainly due to the formation of finer grains as compared to walls built with 20 and 40 s. It can also be observed that the micro-hardness of the WAAM built structure is higher than conventionally built SS 316L (170 HV) at all process conditions [20]. This is due to the finer grain structure that is generally observed in WAAM built samples than compared to conventionally processed samples.

Compression studies are carried out to investigate the impact of interlayer delay on the compressive characteristics of the wall structures (Figure 8). It is observed that higher strength values are observed for walls built with an interlayer delay of 80 s samples. This is mainly due to the finer grain structure observed in the walls built with higher interlayer delay. In addition, the higher density of the walls built with higher interlay delay also influences the strength values [34,35,36,37].

### 4.6. Thermal Analysis

In order to understand the effect of interlayer delay on the preheat temperature of the previously built layer; finite element analysis is carried out. Figure 9 presents the typical temperature distribution obtained from the simulation. By default, Abaqus/CAE displays contour values using the Rainbow color spectrum, which ranges from red (for the highest value 1500 °C) to blue (for the lowest value 1000 °C) (for the minimum value). The magnitude variation of the model is indicated by the various colors of the final output.

Figure 10 presents the effect of interlayer delay on the preheat temperature at different values of interlayer delay. The shorter interlayer delay time results in higher preheat temperature, which has a preheat effect on the deposition of the subsequence layers. In addition, as the deposition moves from the bottom layer to the top layer, the preheat temperature at a particular delay time also increases. This is mainly due to the rapid heat dissipation at the lower layers due to the substrate effect. As the deposition moves to the top layer, heat accumulation increases and the heat dissipation rate reduces. As the deposition proceeds from lower to higher layers, the width of the wall increases, which is caused by the molten metal’s viscosity changing during the deposition since viscosity is temperature dependent. As the temperature of the surrounding atmosphere rises, the viscosity of the molten metal reduces. Thus, at higher layers and lower interlayer delay, the outward flow of the melt pool will be higher resulting in larger width values. The variation in the microstructure and mechanical properties can also be explained with the help of the preheat temperature values presented in Figure 10. The lower preheat temperature at higher interlayer delay results in higher thermal gradients and higher cooling rates, which further results in finer grain structure and higher mechanical properties at the higher interlayer delay [13]. The simulated temperature is compared with experimentally obtained temperature values and it is provided in form of an error bar, as shown in Figure 10.

## 5. Conclusions

The SS 316L wall structures are built at different values of interlayer delay using WAAM and the following conclusions can be derived:Density improvement is observed for higher delay sample and the wall width and height reduced with an increase in the interlayer delay due to reduction in the melt pool temperature;Grain refinement is observed with an increase in the interlayer delay due to a higher cooling rate and lower temperature gradient;An improvement in the micro-hardness and compression strength is observed with an increase in interlayer delay due to fine grain structure;The numerical simulation indicates that the preheat temperature reduces with an increase in the interlayer delay, which results in variations in the geometry, microstructure and mechanical properties.

The study will be further extended to understand the variation of elemental composition from the wire to the samples built at different interlayer delay. Furthermore, Laboratory Equipment Company (LECO) analyses will be carried out to determine the amount of Carbon after WAAM processing. The study will pave a path to build highly dense SS 316L structures using WAAM with controlled geometry for potential applications in building high-quality engineering components.

## Figures and Tables

**Figure 1 materials-14-04187-f001:**
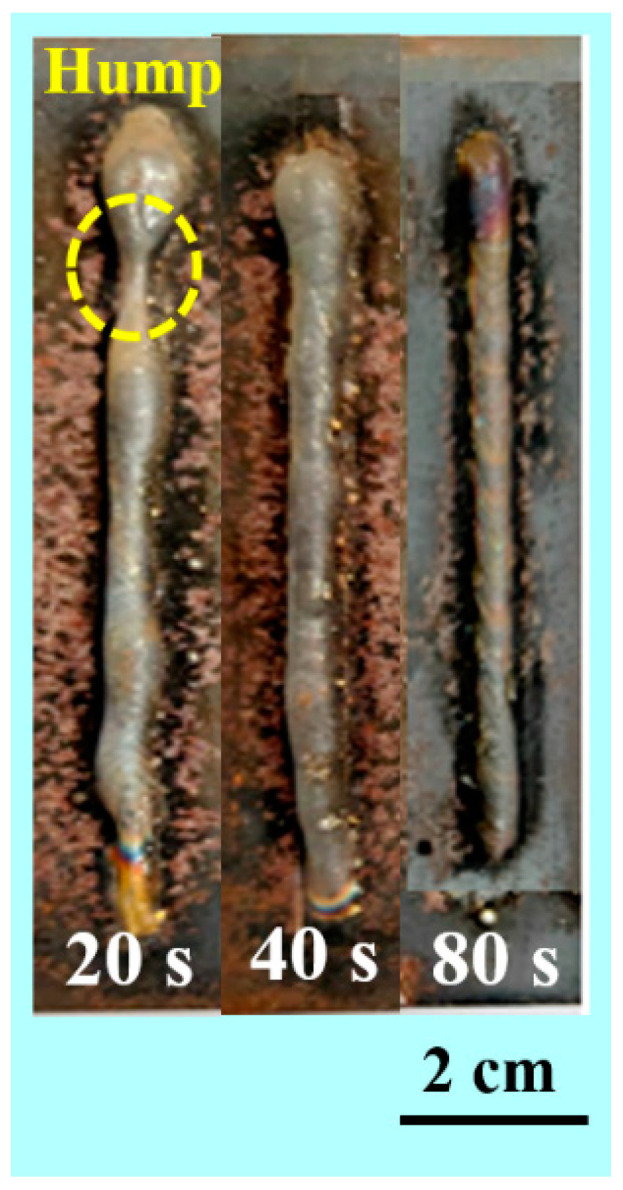
WAAM deposited walls with different thermal delay.

**Figure 2 materials-14-04187-f002:**
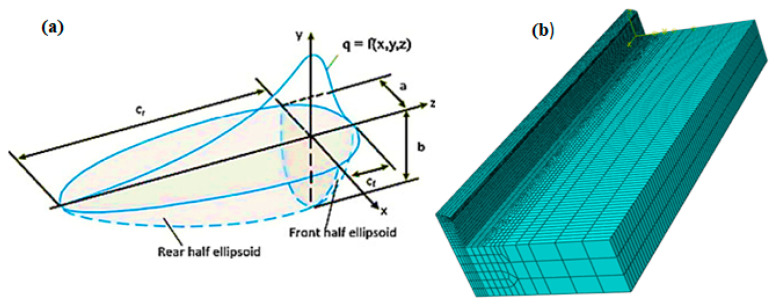
(**a**) Schematics of the Goldak double ellipsoid model and (**b**) meshed WAAM model.

**Figure 3 materials-14-04187-f003:**
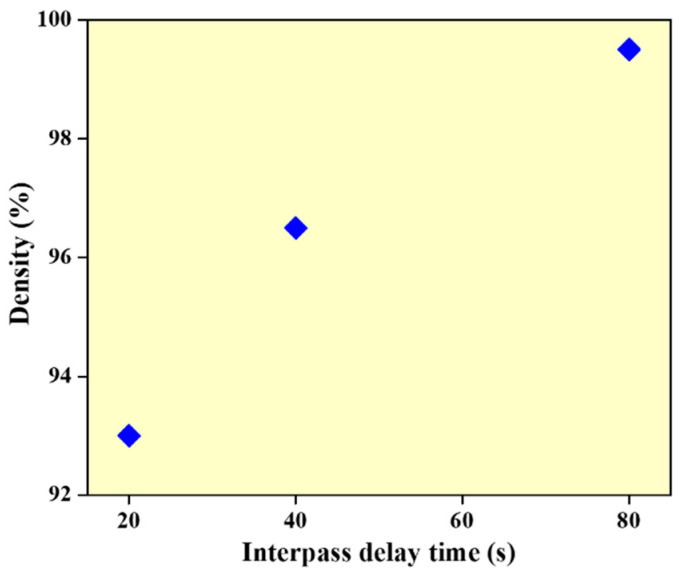
Plot showing the density values of the samples with different thermal delay.

**Figure 4 materials-14-04187-f004:**
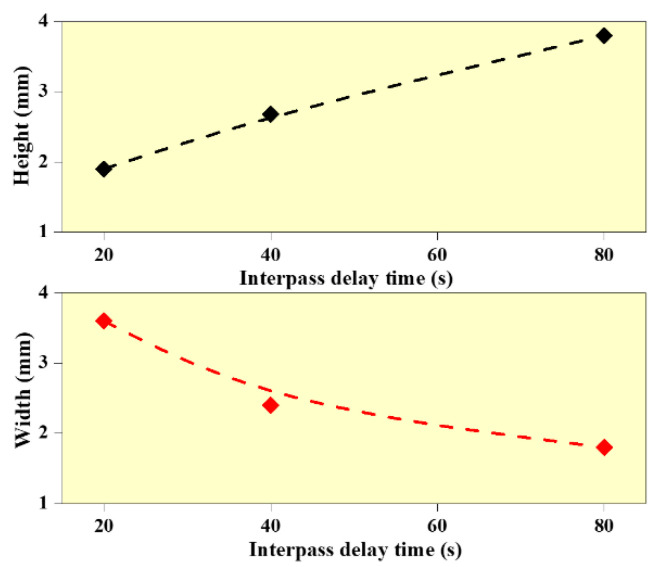
Effect of interlayer delay on wall height and wall width.

**Figure 5 materials-14-04187-f005:**
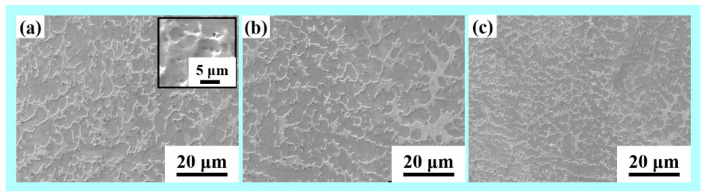
SEM morphology of WAAM samples (**a**) with 20 s (inset: higher magnification image), (**b**) 40 s and (**c**) 80 s interlayer delay.

**Figure 6 materials-14-04187-f006:**
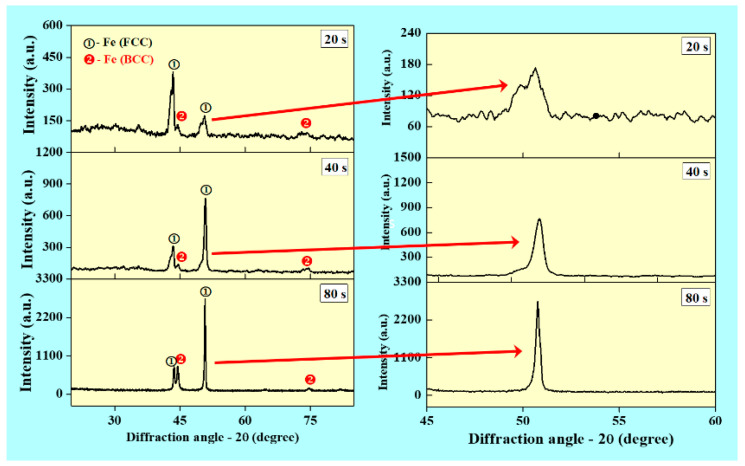
X-ray diffraction (XRD) patterns showing the influence of interlayer delay w.r.t. the phase formation.

**Figure 7 materials-14-04187-f007:**
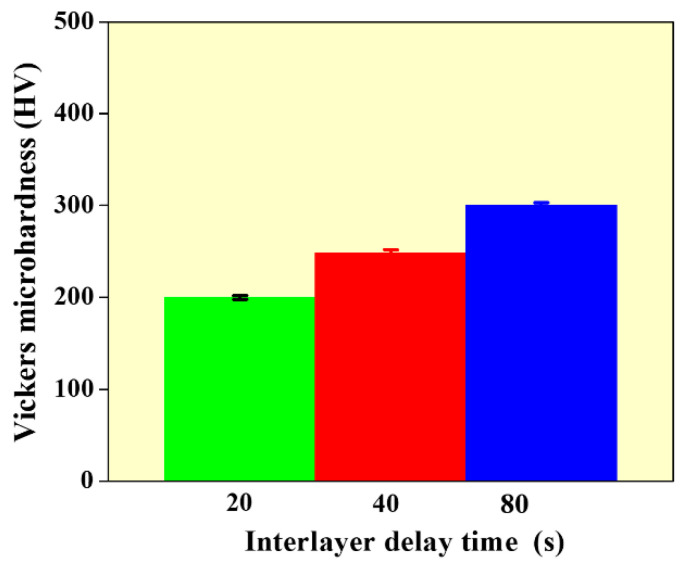
Vickers micro-hardness data as a function of interlayer delay.

**Figure 8 materials-14-04187-f008:**
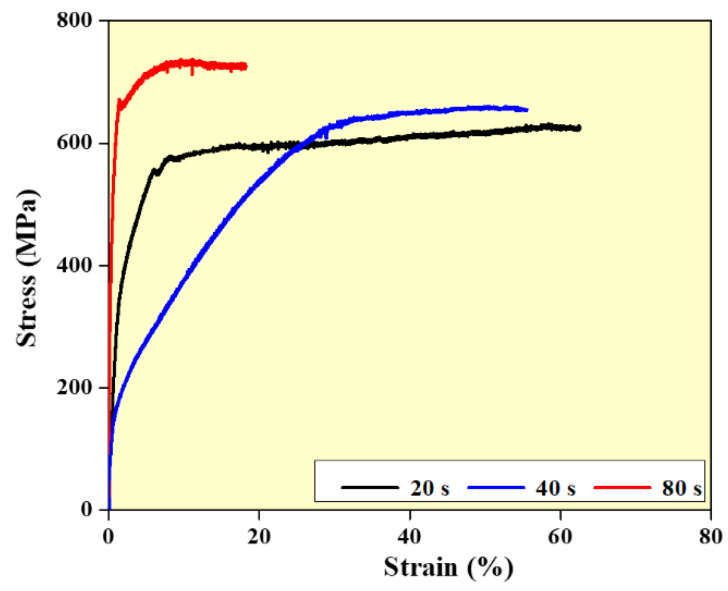
Compression behavior of SS 316L as a function of interlayer delay.

**Figure 9 materials-14-04187-f009:**
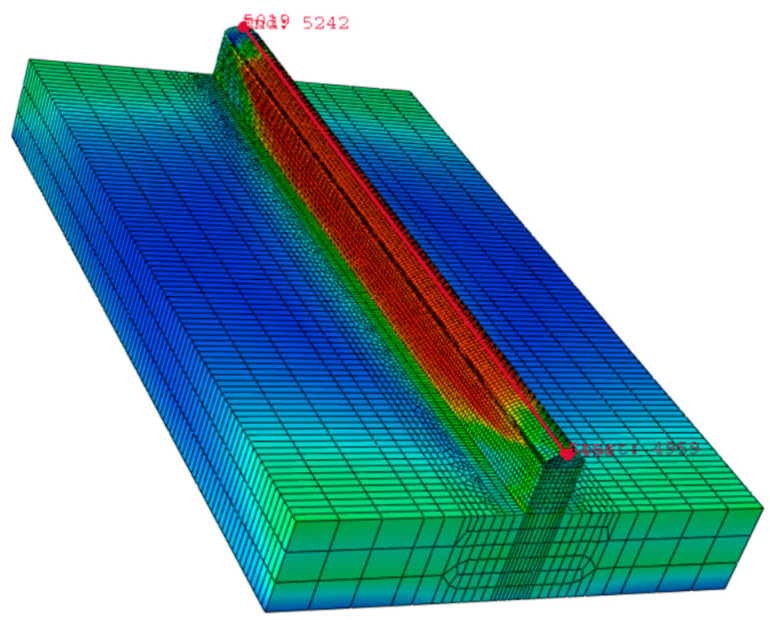
Schematics illustrating the heat distribution in a WAAM process.

**Figure 10 materials-14-04187-f010:**
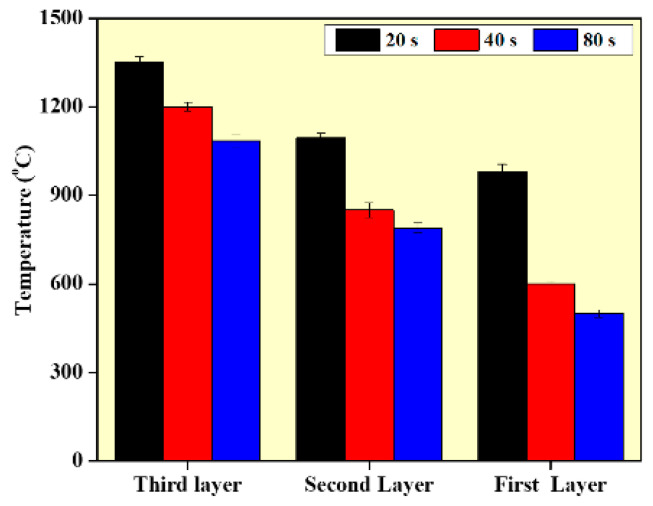
Temperature distribution in the WAAM samples according to the model.

**Table 1 materials-14-04187-t001:** Chemical composition of the SS 316L wire (in wt.%).

Elements	Composition (wt.%)
C	≤0.03
Cr	18.0–20.0
Ni	11.0–14.0
Mo	2.0–3.0
Mn	1.0–2.5
Si	0.30–0.65
P	≤0.03
S	≤0.03
Cu	≤0.75
Fe	Bal.

**Table 2 materials-14-04187-t002:** Optimized parameters for the Wire Arc Additive Manufacturing (WAAM) deposition.

Wire Feed Rate (m/min)	Argon Gas Flow Rate (L/min)	Voltage (V)	Current (m/min)	Scan Speed (mm/s)	Stand of Distance (mm)
5	20	16.5	197	8.57	20

**Table 3 materials-14-04187-t003:** Materials properties used for simulation.

Material Parameter	Symbol	Magnitude
Ablation Temperature (K)	T	1773
Thermal Conductivity (W/m·K)	k	15
Density (kg/m^3^)	ρ	7500
Heat capacity at constant pressure (J/kg·K)	C_p_	468

## Data Availability

The data are a part of ongoing studies and can be made available upon reasonable request from the corresponding author.

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
