# Peer review of "Effect of Interlayer Delay on the Microstructure and Mechanical Properties of Wire Arc Additive Manufactured Wall Structures"

_materials, 2021, doi:10.3390/ma14154187_

Round 1

Reviewer 1 Report

In this study, the effect of interlayer delay on the density, geometry, microstructure and mechanical properties is investigated by experiments and simulation. In general, this manuscript is in good shape. It is deserved to be published after minor revision.

1. line 150: Citation for “  the ‘‘element birth technique"” as well as a brief introduction of this technique is needed.

2.line 169: A blank is missing in “yaxis”

3. The details of the “Scanning Electron Microscopy” should be provided. Type?

4. Figure 6: A additional weak peak can be observed in the magnified view of the 200s plot? Why? Does a new phase present?

5. How were the error bars in Figure 7 determined?

Author Response

In this study, the effect of interlayer delay on the density, geometry, microstructure and mechanical properties is investigated by experiments and simulation. In general, this manuscript is in good shape. It is deserved to be published after minor revision.

Authors would like to thank the reviewer for the appreciation and suggesting the manuscript for publication after minor revision. We have now modified the manuscript as per the advice, which helped us to improve the quality of manuscript.

Comment 1 .  line 150: Citation for “  the ‘‘element birth technique"” as well as a brief introduction of this technique is needed.

Response 1: The authors thank the reviewer for the valuable suggestion and a brief introduction of the technique is now provided in the revised manuscript as: “In this technique, material properties are progressively switched from quiet to active values according to the deposition process as it requires specific elements activation techniques [36] in simulating material deposition. The basic principle is that the quiet material have an extremely low thermal conductivity, hence filler elements in quiet state will experience a significant increase in temperature only if they are directly heated by the external power source. In this work activation start and end temperature are set according to material liquidus and solidus temperatures, respectively [36].”  Citation is also provided in the revised manuscript as per the advice.

Comment 2.line 169: A blank is missing in “yaxis”.

Response 2: Thanks for correction and the blank is now provided in the revised manuscript.

Comment 3. The details of the “Scanning Electron Microscopy” should be provided. Type?

Response 3: The details of Scanning Electron Microscopy are now provided in the Materials and method section of the revised manuscript as For microstructural examination, a scanning electron microscope (SEM) (Make & Model: S-4800 Hitachi) is employed. The secondary electrons (SE) detector was used for obtaining the SEM images [31].”

Comment 4. Figure 6: A additional weak peak can be observed in the magnified view of the 200s plot? Why? Does a new phase present?

Response 4:.Thanks for the critical comment .Small weak peak observed in the XRD analysis belongs to the ferrite phase. This observation is in line with the observations from literature as provided below.

Dadfar, M., Fathi, M. H., Karimzadeh, F., Dadfar, M. R., & Saatchi, A. (2007). Effect of TIG welding on corrosion behavior of 316L stainless steel. Materials Letters, 61(11-12), 2343-2346.

It is provided in the manuscript as “It shows the presence of bimodal microstructure with both austenitic and ferritic phases coexisting at the same time”

Comment 5. How were the error bars in Figure 7 determined?

Response 5: The hardness values are measured at 10 different locations and average value is plotted as Figure 7. The variation in the hardness value is now plotted in the form of error bar.

Reviewer 2 Report

Don’t use abbreviation in the abstract

Page 5:

This does not seem to be a symmetric Gaussian model. At least there is a skew in the model. Authors have mentioned that this is a gaussian model. Please clarify. Also, authors need to specifically explain what each symbol means in the equations 1-5.

Pape 5, Eq 5:

Much information is missing from modeling. Materials properties.

materials constitutive and thermo-physical behavior. model geometry, mesh size and mesh density. Authors need to pay close attention to the modeling section. Also, many people have investigated the thermal modeling and authors need to provide references to the following:

https://link.springer.com/article/10.1007/s11837-019-03872-3

https://www.sciencedirect.com/science/article/pii/S2351978915010124

https://books.google.com/books/about/Additive_Manufacturing_of_Metals.html?id=Pe93zQEACAAJ&source=kp_book_description

Thank you 

Author Response

Comment 1: Don’t use abbreviation in the abstract

Response 1: Thanks for the valuable comment. The abstract is now revised by removing abbreviations.

Comment 2: Page 5:This does not seem to be a symmetric Gaussian model. At least there is a skew in the model. Authors have mentioned that this is a gaussian model. Please clarify. Also, authors need to specifically explain what each symbol means in the equations 1-5.

Response 2:  Authors agree with the reviewer and apologize for the mistake. The GOLDAK heat source model is developed for this numerical approach. Goldak model is a double ellipsoid model which is accurate and flexible to change its size and shape easily for modelling the heat source for WAAM process. All the symbols are now clearly explained in the nomenclature section.

Comment 3: Pape 5, Eq 5: Much information is missing from modeling. Materials properties. materials constitutive and thermo-physical behavior. model geometry, mesh size and mesh density. Authors need to pay close attention to the modeling section. Also, many people have investigated the thermal modeling and authors need to provide references to the following:

https://link.springer.com/article/10.1007/s11837-019-03872-3

https://www.sciencedirect.com/science/article/pii/S2351978915010124

https://books.google.com/books/about/Additive_Manufacturing_of_Metals.html?id=Pe93zQEACAAJ&source=kp_book_description

Response 3: The details pertaining to materials properties, materials constitutive and thermo-physical behavior, model geometry, mesh size, and mesh density are now provided in the “section 3 : numerical modeling” of the revised manuscript. As advised, the references are now updated in the revised manuscript by citing the suggested publications (Ref 33, 34, 35).

Reviewer 3 Report

The paper is focused on a process of growing interest such as WAAM and could be of relevant impact on the relative literature.

However, despite proposing a model for the prediction and analysis of the WAAM process, particularly in terms of the final properties, the paper has flaws, some of which rather severe, that compromise its publishing.

Line 59: WAAM being a multi-physics process, understanding the microstructure of the  deposit, which is directly connected to its mechanical characteristics, is important to maximize the benefits of WAAM.

This sentence does not have sense

Line 118: Table 1 presents the chemical composition of SS 316L wire used for deposition

Table 1 is definetely NOT presenting that!

Line 127: Figure 1 presents the photographic view of the wall structures.

Figure 1 shows quite a not-on-focus image from which no indications could be derived. Moreover, there is no explanation about what is the importance of showing this image.

Line 187:

What are those points, close to equation (1) meaning?

Line 217: Figure 3:

Totally useless...not even providing the right scale on the x-axis

Line 240: Figure. 5 presents the microstructure of the built wall structures obtained using Scanning Electron Microscopy. The microstructure is mainly a mixture of Pearlite and Ferrite phases with different grain sizes and structures.

The only thing which can be derived from Fig. 5 is the rather large amount of scratches in the microstructure!!! It is definetely NOT showing any defined microstructure! Moreover, how is it possible to state that the microstructure is made up of PERLITE and FERRITE? 316L is an austenitic grade... Have been LECO analyses carried out to determine the amount of C after WAAM processing? Why lines fro 250 to 257 provide an explaination which is not involving perlite (i.e: eutectoid transformation of gamma into alfa + Fe3C) if that, as stated in line 241, is the main microstructure present?

Line 276: Figure 7 presents the average micro-hardness of the WAAM built wall structures with different values of interlayer delay. It can be seen that the microhardness increased from 350 HV to 460 HV with an increase in interlayer delay.

Actually the values reported in the charts are NOT those described...

Line 288: This is mainly due to the finer grain structure seen in the walls built with higher interlayer delay.

Where can you see this? There are not microstructures showing grain sizes for the different investigated conditions...

Line 304 and following:

How has the validity of  the model been checked and validated?

Author Response

The paper is focused on a process of growing interest such as WAAM and could be of relevant impact on the relative literature. However, despite proposing a model for the prediction and analysis of the WAAM process, particularly in terms of the final properties, the paper has flaws, some of which rather severe, that compromise its publishing.

Thank you so much for your advice and suggestions. The manuscript is now revised as per the suggestions and the authors believe that the revised manuscript will be suitable for publication.

Comment 1: Line 59: WAAM being a multi-physics process, understanding the microstructure of the deposit, which is directly connected to its mechanical characteristics, is important to maximize the benefits of WAAM.This sentence does not have sense.

Response 1: Thanks for the critical comment. The line is now modified as: “Being a multi-physics phenomenon, there are several factors that controls the microstructure evolution during WAAM. Further, it is critical to understand the co-relation between the microstructure and mechanical properties, which aids in process selection for various applications”

Comment 2 : Line 118: Table 1 presents the chemical composition of SS 316L wire used for deposition. Table 1 is definetely NOT presenting that!

Response 2: The authors thank the reviewer for the valuable suggestion and the Table 1 is now modified in the revised manuscript.

Comment 3: Line 127: Figure 1 presents the photographic view of the wall structures. Figure 1 shows quite a not-on-focus image from which no indications could be derived. Moreover, there is no explanation about what is the importance of showing this image.

Response 3: As per the reviewer suggestions Figure 1 is now replaced with a labelled diagram in the revised manuscript. The explanation is now included in the revised manuscript as “It indicates the presence of humps and spatters during WAAM deposition. In addition, it can be seen that the wall structure built with 20 s interlayer delay period is non-uniform due to heat accumulation effect, while the wall structures built with 40 s and 80 s interlayer delay are uniform.”

Comment 4: Line 187: What are those points, close to equation (1) meaning?

Response 4: The authors apologize for the mistake and the points are now removed from the revised manuscript.

Comment 5: Line 217: Figure 3: Totally useless...not even providing the right scale on the x-axis.

Response 5: The authors apologize for the mistake and the scale is now modified in the revised manuscript.

 Comment 6: Line 240: Figure. 5 presents the microstructure of the built wall structures obtained using Scanning Electron Microscopy. The microstructure is mainly a mixture of Pearlite and Ferrite phases with different grain sizes and structures. The only thing which can be derived from Fig. 5 is the rather large amount of scratches in the microstructure!!! It is definetely NOT showing any defined microstructure! Moreover, how is it possible to state that the microstructure is made up of PERLITE and FERRITE? 316L is an austenitic grade... Have been LECO analyses carried out to determine the amount of C after WAAM processing? Why lines from 250 to 257 provide an explaination which is not involving perlite (i.e: eutectoid transformation of gamma into alfa + Fe3C) if that, as stated in line 241, is the main microstructure present?

Response 6: As per the reviewer suggestion, figure 5 is now replaced with high quality microstructure.  The explanation for microstructural analysis is now modified in the revised manuscript as Figure. 5 presents the microstructure of the built wall structures obtained using Scanning Electron Microscopy. The temperature range of interest for stainless steel is 800°C to 500°C, as this is where the majority of the microstructural changes occur. This is the temperature range where austenite undergoes its solid-state transformation. As a result, in WAAM developed micrographs, the interlayer delay duration is critical. The dark gray region δ phase exhibit fine vermicular morphology within the light gray γ phase matrix. The second δ phase is embedded in the γ matrix, and a large number of small holes (the darker region with circular structure) were also distributed in the matrix due to the high interpass temperature because of low interlayer delay as shown in Figure 5 (a). A part of δ phase re-dissolves in γ phase and retained δ phase exhibits vermicular morphology owing to the effects of subsequent thermal cycles. A large number of equiaxed grains and cellular dendrites were formed and grain size was more for lower interpass delay sample (20 s). The ferrite exhibits reticular morphology within the austenitic. Walls built with higher thermal delay shows finer grains as compared to walls built with lower thermal delay samples, which is mainly due to an increase in the cooling rate with an increase in delay time. With an increase in interlayer delay time, a reduction in the preheat temperature on the surface of the previously built layer is observed, which leads to an increase in the thermal gradient and cooling rates. [13,32]”.

LECO analysis is not performed as the facility is not accessible by the authors at present. The authors are sorry for the above. We understand the importance of the analysis and it will be carried out as a future work. It is now included as future scope at the end of conclusion as “The study will be further extended to understand the variation of elemental composition from the wire to the samples built at different interlayer delay. Further, LECO analyses will be carried out to determine the amount of Carbon after WAAM processing.”

 Comment 7:  Line 276: Figure 7 presents the average micro-hardness of the WAAM built wall structures with different values of interlayer delay. It can be seen that the microhardness increased from 350 HV to 460 HV with an increase in interlayer delay. Actually, the values reported in the charts are NOT those described...

Response 7: Thanks for pointing out this mistake and it is now corrected in the revised manuscript.

Comment 8: Line 288: This is mainly due to the finer grain structure seen in the walls built with higher interlayer delay. Where can you see this? There are not microstructures showing grain sizes for the different investigated conditions...

Response 8: Thanks for this comment. SEM images validating the statement is now included as Figure 5 in the revised manuscript. It is clearly visible that the microstructure is finer for samples built with higher thermal delay (80 s).

Comment 9: Line 304 and following: How has the validity of the model been checked and validated?

Response 9: Thanks for pointing out this mistake and apologies for not adding the experimental values in the original manuscript. The temperature values are measured using a pyrometer and the recorded values are compared with simulated values in the form of error bar as shown in Figure 10. The specifications of pyrometer is now provided in section 2 of the revised manuscript as “An one-colour pyrometer Sensortherm METIS M318 (Sensortherm GmbH, Sulzbach, Germany with a temperature range and spectral range of 150 to 1200 °C and 1.65–2.1 μm, respectively) was used for measuring the temperature on the surface of the previously deposited layer at different interlayer delay conditions.”

Round 2

Reviewer 2 Report

N/A

Reviewer 3 Report

Revisions applied to the paper satisfy the criticisms that were present.

No further changes are thought to be needed.